# The Molecular Chaperone Mechanism of the C-Terminal Domain of Large-Size Subunit Catalases

**DOI:** 10.3390/antiox12040839

**Published:** 2023-03-30

**Authors:** Teresa Nava-Ramírez, Sammy Gutiérrez-Terrazas, Wilhelm Hansberg

**Affiliations:** Departamento de Biología Celular y del Desarrollo, Instituto de Fisiología Celular, Universidad Nacional Autónoma de México—UNAM, Ciudad de México 04510, México; tnava@ifc.unam.mx (T.N.-R.);

**Keywords:** large-size subunit catalase, C-terminal domain, unfolding enzymes, molecular chaperones, symmetric dimers, chaperone function of hydrophobic and charged amino acid residues

## Abstract

Large-size subunit catalases (LSCs) have an additional C-terminal domain (CT) that is structurally similar to Hsp31 and DJ-1 proteins, which have molecular chaperone activity. The CT of LSCs derives from a bacterial Hsp31 protein. There are two CT dimers with inverted symmetry in LSCs, one dimer in each pole of the homotetrameric structure. We previously demonstrated the molecular chaperone activity of the CT of LSCs. Like other chaperones, LSCs are abundant proteins that are induced under stress conditions and during cell differentiation in bacteria and fungi. Here, we analyze the mechanism of the CT of LSCs as an unfolding enzyme. The dimeric form of catalase-3 (CAT-3) CT (TDC3) of *Neurospora crassa* presented the highest activity as compared to its monomeric form. A variant of the CAT-3 CT lacking the last 17 amino acid residues (TDC3^Δ17aa^), a loop containing hydrophobic and charged amino acid residues only, lost most of its unfolding activity. Substituting charged for hydrophobic residues or vice versa in this C-terminal loop diminished the molecular chaperone activity in all the mutant variants analyzed, indicating that these amino acid residues play a relevant role in its unfolding activity. These data suggest that the general unfolding mechanism of CAT-3 CT involves a dimer with an inverted symmetry, and hydrophobic and charged amino acid residues. Each tetramer has four sites of interaction with partially unfolded or misfolded proteins. LSCs preserve their catalase activity under different stress conditions and, at the same time, function as unfolding enzymes.

## 1. Introduction

Proteins can have several metastable states and many cellular proteins become partially denatured or acquire a non-functional conformation particularly under stressful conditions. Molecular chaperones are proteins that assist other proteins to gain their native functional conformation which usually is the system’s state of least energy. Molecular chaperones are unfolding enzymes—this is probably the unifying function of these proteins, and not, or not mainly, preventing aggregation of partially unfolded or misfolded proteins, as has been emphasized in the scientific literature [1,2]. Some unfolding enzymes/molecular chaperones are disaggregases, indicating that the unfolding activity is also critical for protein disaggregation [3].

Unfolding enzymes are essential to all cells and are present in many if not all cellular compartments, the extracellular space, and the bacterial periplasm [4,5,6,7,8,9,10]. They are abundant proteins [11], nevertheless the expression of many of these proteins, but not all, increases under different stress conditions [12,13]. Expression of numerous chaperones rises during heat shock and thus many unfolding enzymes are called heat shock proteins (Hsp). Unfolding enzymes can have overlapping functions and usually low specificity in their interactions with partially unfolded or misfolded proteins and they often display cooperativity in their interactions with numerous proteins [14,15,16]. Some are large and complex molecular chaperones, such as Hsp60 (GroEL/ES), Hsp70/Hsp110 (DnaK), Hsp104 (ClpB), and Hsp90 (HtpG), that utilize ATP to attain a structural rearrangement and are usually assisted by co-chaperones. ATP-fueled chaperones drive their substrate proteins out of equilibrium, maximizing the nonequilibrium native yield in a given time rather than the absolute yield or folding rate [2]. Other chaperones are low-molecular proteins and do not require ATP for their activity, such as the small Hsp (sHsp), the DJ-1 superfamily proteins, and other small-size chaperones, but they all share a similar molecular mechanism. Many of the low-molecular unfolding enzymes function as homodimers, although some of them can form higher-order oligomers [17,18]. Interactions of molecular chaperons with a partially unfolded or misfolded protein are very dynamic and have been studied using NMR spectroscopy [19,20].

Monofunctional heme-catalases constitute a monophyletic group that is divided into two classes: LSCs and small-size subunit catalases (SSCs) [21,22,23,24]. LSCs are similar to SSCs but have an additional mobile coil (MC) (29–35 amino acids) and a CT (150–190 amino acids). LSCs are present in bacteria and filamentous fungi but absent in plants and animals, while SSCs are present in all phyla [21]. The CT of LSCs is conserved in its structure but not in its primary sequence and has an amino acid composition like very stable proteins [25]. The CT has been assigned to the DJ-1/PfpI superfamily [26] and the class I glutamine amidotransferase-like superfamily [27].

*N. crassa* asexual spore formation is a response to hyperoxidant conditions [28,29,30]. The two LSCs of *N. crassa*, CAT-3 (type L2) and catalase-1 (CAT-1) (type L1) [23], are highly induced during the cell-differentiation process; CAT-3 at the start, when hyphae adhere to each other, and CAT-1 at the end, when conidia are made [31,32].

The structure of the CT of LSCs is similar to Hsp31 and DJ-1 proteins. The origin of the CT is traced to a bacterial Hsp31—the MC plus the CT (MC_CT) of bacterial and fungal LSCs have higher similarity with bacterial HchA (Hsp31) sequences than any other protein of the DJ-1/PfpI superfamily. This is mainly because the N-terminal of bacterial HchA is similar to the MC [25]. From 186 sequences, 30 bacterial HchA sequences from different phyla consistently aligned with a file of bacterial and fungal MC_CT sequences [25]. Because the related bacterial Hsp31 sequences came from many different phyla, it was concluded that the likely fusion between an SSC and a bacterial Hsp31 occurred very early in the phylogeny of bacteria before the diversification of extant phyla. A structural alignment with the closest bacterial sequences showed that structures are indeed very similar to the CT; one of them gave an RMSD of 2.2 Å [24].

The CT of LSCs has an unfolding activity that can act on other proteins, reverting partially denatured or misfolded proteins to the active conformation. LSCs or TDC3 prevent denaturation of alcohol dehydrogenase (ADH) or other proteins by heat, urea, or H_2_O_2_, whereas SSCs, CAT-3 without the CT (C3^ΔTD^), and CAT-1 lacking CT (C63) do not have this effect [33]. Similar results are obtained if ADH is previously denatured by heat and then a LSCs or the TDC3 is added. The TDC3 also protects both the C3^ΔTD^ and the bovine liver catalase from heat denaturation [33]. The unfolding activity of CAT-3 and TDC3 increases the survival of *E. coli* under different stress conditions whereas the C3^ΔTD^ does not [33].

Here, we analyzed the molecular chaperone mechanism of the CT of CAT-3. We found that the TDC3 acts mainly as a dimer and that hydrophobic and charged amino acid residues participate in its unfolding activity. CAT-3 or TDC3 lacking the last 17 amino acid residues (CAT-3^Δ17aa^ and TDC3^Δ17aa^) lost most of its unfolding activity. Switching charged for hydrophobic residues or vice versa in this C-terminal loop considerably diminished its molecular chaperone activity. It was concluded that the unfolding mechanisms of the TDC3 is associated with its dimeric conformation and the presence of several hydrophobic and charged residues, many of them localized in the C-terminal loop.

## 2. Materials and Methods

### 2.1. Proteins and Reagents

The following enzymes were obtained from Sigma-Aldrich, Saint Louis, MO, USA: catalase CAT-A (C3515) from *Saccharomyces cerevisiae*, bovine liver catalase (C9322), ADH from *S. cerevisiae* (9031-72-5), bovine serum albumin (BSA) (A7906), and subtilisin (P5380). Other reagents used were: isopropyl β-D-thiogalactopyranoside (IPTG) (NZYTech, Lisboa Portugal, MB026), 4,4′-dianilino-1,1′-binaphthyl-5,5′-disulfonic acid, dipotassium salt (bis-ANS) (Thermo Scientific, Alvarado, TX, USA, B153), LB medium (Sigma-Aldrich L3022), and hemin (Frontier Scientific, Newark, NJ, USA, H651-9).

### 2.2. Plasmid Constructs and Expression in E. coli

The catalase domain C3^∆TD^ (residues 1–518) and the TDC3 (residues 568–719) were cloned into the pCold^TM^ I plasmid as described in [33]. The Hsp31 gene was obtained from the DNA of *E. coli* by amplification with specific oligonucleotides and also cloned into the multicloning site of pCold I as described in [33].

CAT-3^Δ17aa^ and TDC3^Δ17aa^ variants were amplified from the *cat-3* gene without introns, using oligonucleotides designed to eliminate the carboxy-terminal region. The variants were introduced into a pCold I plasmid using the *Xba*I restriction site as was done for the complete variants. The single mutants TDC3^Q652C^, TDC3^K662L^, TDC3^L702E^, TDC3^F705E^, TDC3^R710W^, TDC3^F711E^, and TDC3^D714I^ were produced by site-directed mutagenesis using the QuickChange II kit (Agilent, Santa Clara, CA, USA, 200524) following the indications of the manufacturer.

From an isolated single colony, bacteria were grown overnight in LB Lennox medium containing 100 mg/mL ampicillin at 37 °C, agitating at 200 rpm. The preculture was diluted 1:10 into fresh LB medium and incubation was continued to an OD at 600 nm of 0.4. Expression of the protein was induced with 1 mM IPTG and incubated at 16 °C, with 200 rpm agitation for 48 h. Hemin, 30 mM, pH 9.6, was added to the medium when catalases were expressed.

The expressed protein, tagged with six histidine residues, was purified with a Ni-agarose (QIAGEN N.V., Germantown, MD, USA, 30250) affinity column following instructions of the dealer (5th edition). Fractions containing most of the protein were concentrated by centrifugation in an Amicon filter (either 30,000 or 10,000 Da cutting size) and verified by PAGE-SDS (either 8% or 15% acrylamide) and stained with Coomassie Brilliant Blue.

### 2.3. Enzyme Purification and Analysis

CAT-1 was purified directly from *N. crassa* conidia following the method described in [32]. The purified CAT-1 was treated with subtilisin at a 7:1 protease/protein ratio at 37 °C for 1 h to give the active C63 catalase [33].

In-gel catalase activity after PAGE (using 1.44% glycine, 0.3% Trizma base, pH 7.0, 2 h at 150 V) was done as described previously [32,33].

### 2.4. TDC Analysis and TDC3 Dimer and Monomer Formation

The molecular weight of the purified TDC3 was determined with a Superdex 75 HR 10/300 column coupled to a FPLC system (GE Healthcare Life Sciences, San Diego, CA, USA) according to the specifications by the company. TDC3 (1 mg/mL) was injected to the column with a flux of 0.75 mL/min. The molecular mass was determined with a mix of gel filtration standards (BioRad, Hercules, CA, USA, 1511901).

To form the cysteine disulfide in TDC3^Q652C^ the protocol described in reference [34] was followed: TDC3^Q652C^ (6 µM) in 100 mM Tris, pH 8.5, was mixed with 10 volumes of 9 M urea (HPLC grade), treated subsequently with DTT (0.5 M), incubated overnight at 4 °C and thereafter dialyzed by passing through a Sephadex G25 column in 0.1 M Tris-HCl buffer, pH 8.0. Then, 160 µM of H_2_O_2_ was added to the reduced TDC3^Q652C^, incubated at 25 °C for one hour, and thereafter dialyzed by passing through a Sephadex G25 column.

For the formation of the TDC3^Q652C^ monomer, the reduced TDC3^Q652C^ in 100 mM Tris, pH 7.2, was mixed with N-ethylmaleimide (NEM) (Sigma-Aldrich 23030) in a ratio 1:1 protein/NEM (6 µM) and incubated at 25 °C for two hours [35]. To derivatize the Cys652 with glutathione, the reduced TDC3^Q652C^ in 100 mM Tris, pH 7.2, was incubated with 1 mM of GSSH (Sigma-Aldrich G4501) at 25 °C for 1 h [36].

Monomer and dimer formation was verified by SDS-PAGE using 15% of acrylamide (Appendix A). Folding of the TDC3 monomer, dimer, and the TDC3 mutant variants was confirmed by near-UV CD spectra (Appendix A). Spectra were run between 200 and 260 nm, at 25 °C, with a protein concentration of 0.2 mg/mL in phosphate buffer 50 mM, pH 7.8, in a Chirascan^TM^ spectropolarimeter (Applied Photophysics^®^, Leatherhead, UK).

### 2.5. Chaperone Assay

ADH (6.2 µM calculated as tetramers) in 50 mM phosphate buffer, pH 7.0, was heat-denatured at 45 °C for 150 min and light scattering at 360 nm was followed in a Beckman Coulter DU-650 spectrophotometer. The control sample contained 6.2 µM BSA instead of a chaperone. CAT-1 and CAT-3 was added at either 1.5 or 3 μM, calculated as tetramers; C63 and C3^∆TD^ at 6 µM, calculated as tetramers; and TDC3 and Hsp31 at 6 μM, calculated as dimers. Mixtures were done at RT in a 500 µL quartz cuvette with a final volume of 400 µL.

### 2.6. Detection of Hydrophobic Regions

A concentration of 0.8 μM of each protein in 50 mM phosphate buffer, pH 7.4, was incubated with bis-ANS (1.2 μM), either at RT or 45 °C, for 30 min. Fluorescence was measured in a BioTek Synergy Mx Microplate Reader, excited at 370 nm, and emission determined between 400 and 600 nm at RT. 

CAT-1, CAT-3 (each at 3 μM), Hsp31, and TDC3 (each at 6 μM) in 50 mM phosphate buffer, pH 7.4, were incubated with bis-ANS and thereafter exposed to UV light 360–370 nm for 10 min at RT. Proteins were dialyzed by passing them through a Sephadex G25 resin and used for the chaperone assay.

### 2.7. Unfolding Activity in the Presence of Increasing Ionic Strength or at Various pH

NaCl at 0, 0.15, 0.3, 0.5, 0.7, and 1 M was added to CAT-3 (3 µM), TDC3 (6 µM), and Hsp31 (6 µM) in 10 mM phosphate buffer, pH 7.0, (with an ionic strength of 0.028 M) [37] and tested on the heat denaturation assay of ADH, as described.

CAT-1, CAT-3, and Hsp31 at 3 and 6 µM were tested in the ADH chaperon assay using either 50 mM sodium phosphate at pH 6.0, 50 mM Na/K phosphate at pH 7.8, or 50 mM sodium borate at pH 9.0.

## 3. Results

### 3.1. Does the TDC3 Function as a Dimer or as a Monomer?

The CT origin from a bacterial Hsp31 [25] and the requirement of the dimeric form for chaperone activity of Hsp31 [17,38], led us to question whether the TDC3 also functions as a dimer. Size-exclusion chromatography showed that TDC3 in solution had monomers and dimers in similar proportions, and a small fraction of aggregates of high molecular weight. The molecular mass of the monomer obtained was 18.24 kDa, very close to the theoretical value 18.02 kDa (calculated by Expasy); while for the dimer, it was 44.61 kDa, approximately 5 kDa higher than the theoretical value of 39.04 kDa (Appendix A).

To evaluate if the TDC3 dimer is required for its activity, the Gln652 was substituted by a cysteine (TDC3^Q652C^) to form a disulfide bond between TDC3 monomers allowing, in this way, the stabilization of the TDC3 dimer. The Q652 is found in the alpha helix located at the center of the interface of the inverted symmetric dimers in the CAT-3 tetrameric structure [39] (PDB 3EJ6). No cysteines are present in the native TDC3. To obtain TDC3 monomers that do not form dimers in solution, the cysteine residue of the Q652C variant was derivatized with NEM (Appendix A). The unfolding activity of the TDC3^Q652C^ dimer was similar to the TDC3 without modification. The TDC3^Q652C^ monomer derivatized with NEM also showed chaperone activity but it increasingly failed to preserve the ADH in its active conformation through the incubation time (Figure 1A,B). When the concentration of the NEM-derivatized TDC3^Q652C^ monomer was increased two times, a complete ADH protection effect was observed (Figure 1A,B). The monomer having the Cys derivatized with glutathione gave similar results (Figure 1A,B).

### 3.2. Hydrophobic Regions Are Involved in the Unfolding Activity of the CT

Exposure of hydrophobic patches at the interacting protein surface of the molecular chaperone is a general mechanism by which unfolding enzymes recognize partially unfolded or misfolded polypeptides. We therefore assayed the bis-ANS probe that fluoresces when it interacts with hydrophobic regions at the protein surface. CAT-1, CAT-3, and TDC3 had a high fluorescence that increased further when incubated at 45 °C; in contrast, the SSCs CAT-A or BSA showed a low fluorescence and a small increase at 45 °C (Figure 2A–C). This indicates that CAT-1, CAT-3, and TDC3 exposed more hydrophobic regions when incubated at 45 °C, similar to Hsp31 (Figure 2A,C) [40].

To assure that the exposed hydrophobic regions that interact with bis-ANS are involved in the unfolding activity, the bis-ANS, by UV light treatment, was covalently bond to its binding residues. Proteins treated with bis-ANS plus UV light lost the ability to inhibit heat denaturation of ADH (Figure 3A,B). The mere presence of bis-ANS, added 10 min after initiation of the reaction, also hampered the unfolding activity (Figure 3C).

### 3.3. Charged Amino Acid Residues Are Involved in the Unfolding Activity of the CT

It has been shown that the charged amino acid residues at the interacting protein surfaces participate in the general mechanism by which molecular chaperones recognize partially unfolded or misfolded polypeptides [37]. To analyze the effect of charged amino acid residues in TDC3, the inhibition of ADH heat denaturation by CAT-3 and TDC3 was done in the presence of different concentrations of NaCl. The increase of the ionic strength led to a reduction in the unfolding activity of CAT-3 and TDC3 (Figure 4).

This result indicates that charged amino acid residues are involved in the protein–protein recognition between the ADH and the unfolding enzymes. The chaperone activity of Hsp31 was more affected than the activity of CAT-3 and TDC3. Remarkably, even in the presence of 1 M NaCl, the CAT-3 and TDC3 preserved two thirds of its unfolding activity. We also assayed the participation of charged amino acid residues by changing the pH of the buffer. The pH had a general low effect on the unfolding activity; the effect was different for each of the enzymes tested (Appendix A).

### 3.4. Identification of the Region Responsible for the Unfolding Activity

To identify those regions involved in the CT unfolding activity, we determined the interacting regions in the dimer formed by the CT at both poles of the tetrameric CAT-3 structure. For this, we obtained the surface electrostatic potential of the TDC3 dimer and the amino acid residues that are exposed to the solvent and that could interact with other proteins (Figure 5A,B).

Many of the predicted amino acid residues are localized in the C-terminal loop and the cleft formed in the interface of the two CTs with inverted symmetry. The C-terminal loop has hydrophobic and charged amino acid residues only. We therefore eliminated the last 17 amino acid residues of CAT-3 and TDC3, consisting of seven hydrophobic and nine charged amino acid residues and one glycine (low hydrophobicity). Of these residues, V704 and F705 have a high probability, and K706, F707, R710, and F711 have intermediate probability of interaction with other proteins. CAT-3 and TDC3 without the C-terminal loop, CAT-3^Δ17aa^ and TDC3^Δ17aa^, lost 85% of their unfolding activity (Figure 5C,D).

To determine which of these amino acid residues participate in the unfolding activity of the TDC3 dimer, we substituted some residues—hydrophobic for charged and charged for hydrophobic. To select substitutions that had a less disturbing effect on the conformation of the domain a theoretical analysis was done using the server Duet [42]. All substitutions showed a clear effect on the TDC3 unfolding activity, between 43 and 50% of the control activity (Figure 6A,B). However, it was possible to distinguish differences in their denaturation rates; rate was relatively fast for the K662L variant (maximal activity in 20 min), intermediate for R710W and F711E (maximal activity at 30 min), and slow for L702E, F705E, and D714I variants (maximal activity at 35–40 min) (Figure 6A,C). Near-UV spectra showed similar amounts of alpha helices in all TDC3 variants with the exception of TDC3^D714^ which had more beta strands and fewer unstructured regions (Appendix A).

## 4. Discussion

Unfolding enzymes are abundant proteins that assist other proteins to attain or maintain their native conformation. They do this essential task by interacting with hydrophobic regions of proteins partially denatured or misfolded in a non-functional conformation. Operating by hydrophobic and electrostatic interaction, unfolding enzymes drive their substrate proteins out of equilibrium allowing the exposed hydrophobic regions of these proteins to acquire their normal internal location and, in this way, recover the native functional conformation.

LSCs are abundant proteins that are essential to reversing increasing H_2_O_2_ concentrations during cellular oxidative stress. The CT of LSCs have an unfolding activity that assists other proteins to acquire or maintain their active conformation and increases the survival of *E. coli* under heat shock or oxidative stress [33]. Using heat denaturation of ADH as a molecular chaperone assay, we investigated the mechanism by which the CT of LSCs functions as an unfolding enzyme.

The CTs are structured as a dimers, one dimer in each pole of the LSC tetrameric structure. The dimer conformation is different from the Hsp31 and DJ-1 dimers, although the monomers are structurally very similar to the CT of LSCs. In the DJ-1/PfpI superfamily, four dimerization modes were described [17]; the CT of LSCs constitutes a fifth mode of dimerization into symmetric dimer [25].

Many low molecular chaperones, such as Spy, alpha-crystallin B, Hsp20, Hsp33, Hsp31, and DJ-1 are active as dimers [17,38,43]. The TDC3 spontaneously forms dimers in solution (Appendix A). Docking experiments also indicated dimer formation of CTs, and some of the dimers obtained by docking had the inverted symmetry found in LSCs (Appendix A). To have homogeneous populations of dimers and monomers, the Q652C variant of TDC3 was obtained to maintain the dimeric structure by forming a disulfide bridge. We showed that the dimer was more active than the monomer, whose formation was induced by derivatizing the cysteine of the Q652C variant with NEM or GSH.

The two CTs with inverted symmetry form a cleft between them with the two C-terminal loop positioned in the extremes of the cleft (Figure 5A,B). In the C-terminal loop and along the cleft, several amino acid residues have a high probability of interaction with other proteins (Figure 5B). The C-terminal loop contains 17 amino acid residues that have a surface potential of −10 kcal/mole (red in Figure 5A). This loop is composed of charged (9) and hydrophobic residues (8) only; six of its residues are predicted to interact with other proteins. We observed a reduction of 85% in the molecular chaperone activity by deleting the 17 C-terminal loop, indicating that this loop, probably together with other amino acid residues in the cleft, is required for the unfolding activity. Substitution of hydrophobic for charged amino acid residues and vice versa in the C-terminal loop of TDC3 resulted in a decrease in the unfolding activity of 43–50% of the control activity. This further indicates that hydrophobic and charged amino acid residues of this loop are important for the unfolding activity of the TDC3. Near-UV CD spectra indicate that the TDC3^D714I^ variant had an increased numbers of beta stands and less unstructured regions, suggesting a beta strand in the loose C-terminal loop (Appendix A), which could account for its slow rate of its chaperone activity.

Increased exposure of hydrophobic residues to temperature in CAT-1, CAT-3, and TDC3 was shown by an increase in the fluorescence of the bis-ANS probe, which involves the hydrophobic amino acid residues. Charged residues are also part of the general mechanism of unfolding enzymes for the recognition of unfolded or misfolded proteins. Increasing the ionic strength with NaCl reduced the ionic interactions and the unfolding activity of CAT-3 and TDC3. Together with the results of substitution of charged residues in the C-terminal loop this indicated that charged residues are important for the unfolding activity of CAT-3 and TDC3.

Interestingly, the SSCs of *Arabidopsis thaliana* [44] and *Oryza sativa* (rice) [45] are stabilized by catalase-specific chaperones, which are essential for the three catalase activities; furthermore, the *Arabidopsis thaliana* SSCs are also protected by a peroxisomal sHsp [46].

We observed general overlapping properties among of antioxidant and unfolding enzymes: both enzymes are abundant proteins, and they are induced under stress conditions and during cell differentiation [23,47]. Upon oxidation, peroxiredoxins are inactivated as peroxidases and acquire chaperone activity with oligomerization [48]. Thioredoxin, glutaredoxins, and disulfide isomerases also function as molecular chaperones [49,50]. In this work, we present further data supporting that the LSCs evolved to have antioxidant and unfolding activities in two separated domains of the protein.

## 5. Conclusions

LSCs are also unfolding enzymes. Molecular chaperone activity is due to an additional CT which is derived from a bacterial Hsp31. In the LSCs structure the CTs are structured as dimers with inverted symmetry in each pole of the tetrameric structure. The CT dimers of CAT-3 have a higher chaperone activity than the monomers. The unfolding enzyme activity of the CT is localized mainly in the C-terminal loop. Deletion of this loop resulted in major loss of TDC3 chaperone activity. Hydrophobic and charged amino acid residues of the C-terminal loop are critical for the chaperone activity. Substitution of hydrophobic for charged amino acid residues and vice versa in this loop considerably decreased the unfolding activity.

## Figures and Tables

**Figure 1 antioxidants-12-00839-f001:**
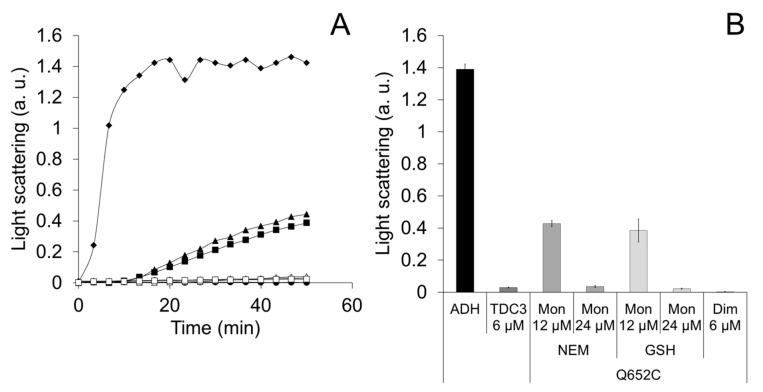
Unfolding activity of a TDC3^Q652C^ dimer and its derivatized monomer with either NEM or GSH. (**A**) Light-scattering representative traces of ADH denaturing by incubation at 45 °C in the presence of TDC3 variants. ADH with BSA (6 µM) (control) (closed rhomboids), TDC3 (6 µM) (open squares), TDC3^Q652C^ disulfide dimer (6 µM) (closed squares), NEM-derivatized TDC3^Q652C^ monomer (12 µM) (close triangles), NEM-derivatized TDC3^Q652C^ monomer (24 µM) (open triangles), glutathione-derivatized TDC3^Q652C^ monomer (12 µM) (closed circles), and glutathione-derivatized TDC3^Q652C^ monomer (24 µM) (open circles). (**B**) Average of three independent assays determined after 50 min.

**Figure 2 antioxidants-12-00839-f002:**
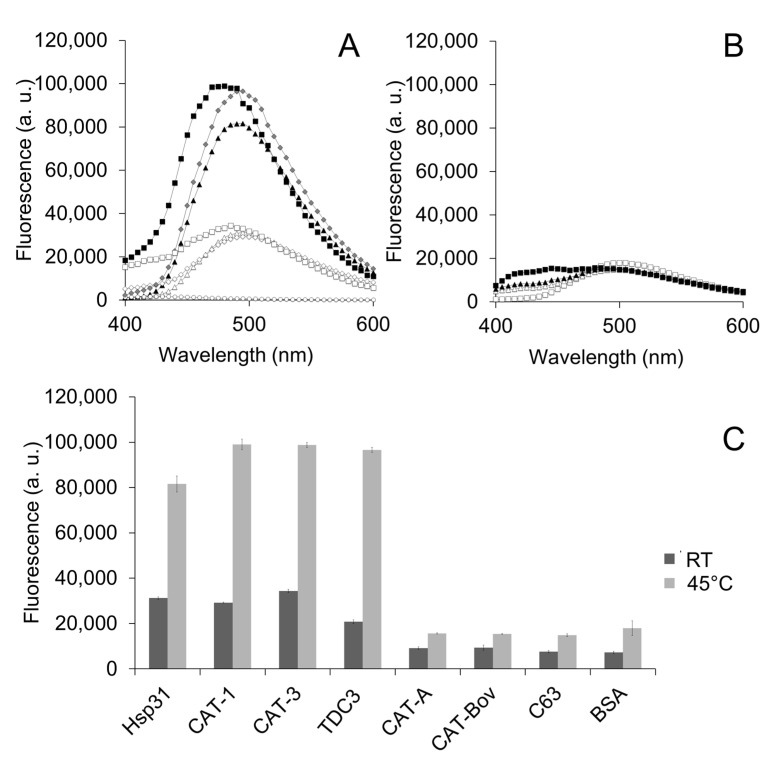
Hydrophobic regions at the CT are exposed by temperature. (**A**) Fluorescence of the bis-ANS (1.2 µM) when it binds to the hydrophobic zones of the protein surface. CAT-1, CAT-3, and TDC3, together with the Hsp31 (each at 0.8 µM) presented fluorescence at RT which increased 3-fold when incubated at 45 °C for 30 min. Hsp31 (open triangles at RT; closed triangles at 45 °C), CAT-3 (open squares at RT; closed squares at 45 °C), and TDC3 (open rhomboids at RT; closed rhomboids at 45 °C). (**B**) The SSC CAT-A and BSA showed a low fluorescence and a small increase with temperature. CAT-A (open squares at RT; closed squares at 45 °C) and BSA (open triangles at RT, closed triangles at 45 °C). (**C**) Average of highest values of three independent experiments.

**Figure 3 antioxidants-12-00839-f003:**
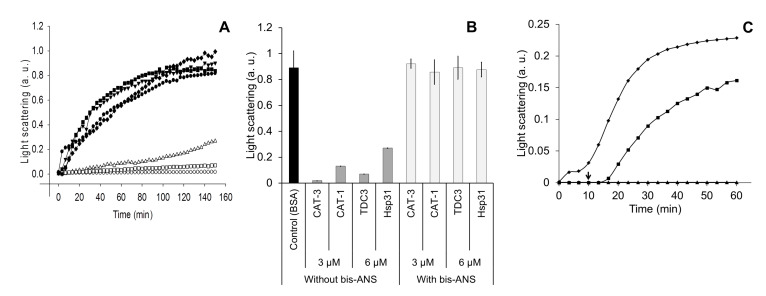
Proteins treated with the bis-ANS probe lost the ability to inhibit heat denaturation of ADH. (**A**) Light scattering of ADH (6 µM) when heat denatured (45 °C) in the presence of chaperones with the bis-ANS incorporated covalently. Hsp31, CAT-1, CAT-3, and TDC3 (3 and 6 µM) were incubated with bis-ANS (1.2 µM) at 45 °C for 30 min and thereafter irradiated with UV light (366 nm) for 10 min at RT. ADH with BSA (6 µM) (closed circles), CAT-3 (3 µM) without bis-ANS (open circles), CAT-3 (3 µM) with incorporated bis-ANS (closed triangles); TDC3 (6 µM) without bis-ANS (open squares), TDC3 (6 µM) with incorporated bis-ANS (closed rhomboids); Hsp31 (6 µM) without bis-ANS (open triangles), and Hsp31 (6 µM) with incorporated bis-ANS (closed squares). (**B**) Average of three independent assays. (**C**) Light scattering of ADH (6 µM) when denatured at 45 °C for 150 min in the presence of BSA (6 µM) (rhomboids), TDC3 (6 µM) (triangles), and TDC3 (6 µM) but adding bis-ANS (1.2 µM) 10 min after starting the reaction (indicated by arrow) (squares). To avoid photo-incorporation of the bis-ANS probe, light scattering was determined at 500 nm in this assay.

**Figure 4 antioxidants-12-00839-f004:**
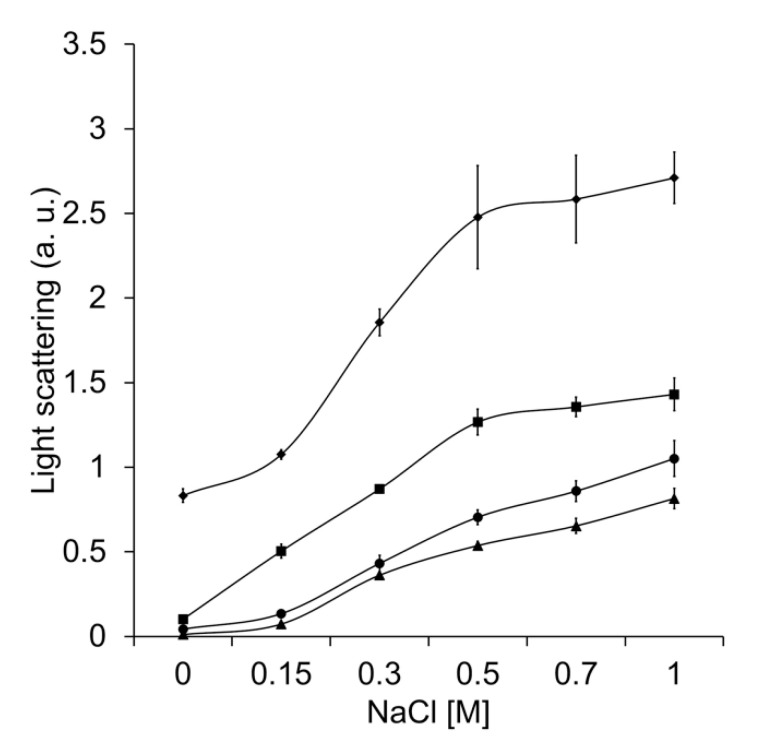
Increased ionic force (NaCl concentration) partially impaired unfolding activity. Heat denaturation of ADH (6 µM), measured by light scattering at 360 nm, in the presence of BSA (6 µM) (rhomboids), Hsp31 (6 µM) (squares), CAT-3 (3 µM) (triangles), and TDC3 (6 µM) (circles) at different NaCl concentrations (0, 0.15, 0.3, 0.5, 0.7, and 1 M) in 10 mM phosphate buffer, pH 7.8. Average of the final light-scattering value from three independent experiments.

**Figure 5 antioxidants-12-00839-f005:**
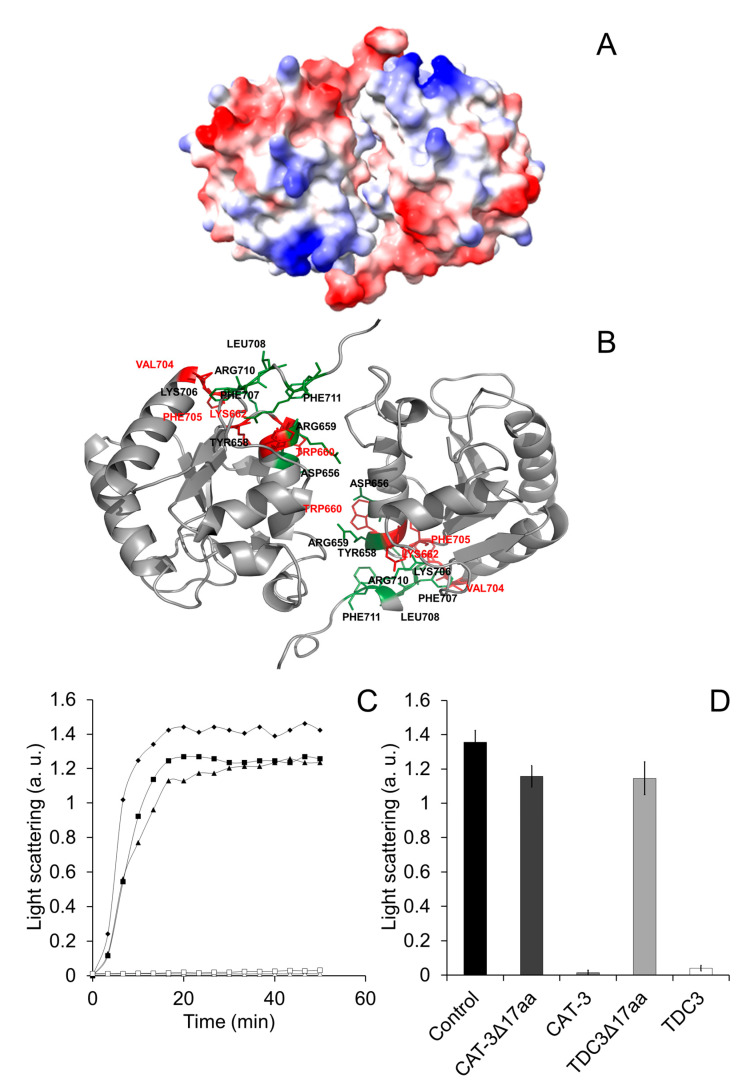
Determination of the amino acid residues in the TDC3 surface that could interact with the substrate proteins. (**A**) Surface representation of the TDC3 dimer with the electrostatic potentials calculated by UCSF Chimera between −10 kcal/mole (red) and +10 kcal/mole (blue). (**B**) Using the InterProSurf server [41], a prediction of the zones of the TDC3 dimer that could interact with other proteins was obtained. Amino acid residues with the highest probability of interaction are highlighted in red (W660, K662, V704, and F705) and those with medium probability are marked in green (D656, Y658, R659, K706, F707, R710, and F711). (**C**) Representative traces for ADH (6 µM) heat denaturation, determined by light scattering at 360 nm, at 50 °C for 50 min, in the presence of: BSA (6 µM) (rhomboids), CAT-3 (3 µM) (open triangles), TDC3 (6 µM) (open squares), CAT-3^Δ17aa^ (3 µM) (filled triangles), and TDC3^Δ17aa^ (6 µM) (filled squares). (**D**) Average of the final light-scattering value from three independent experiments.

**Figure 6 antioxidants-12-00839-f006:**
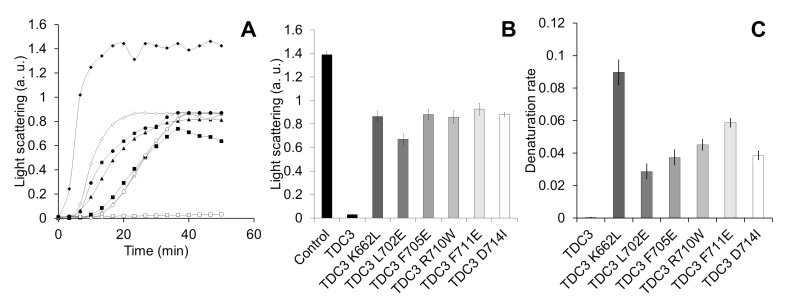
Mutation of specific amino acid residues of the C-terminal loop diminishes chaperone activity. (**A**) Representative traces for ADH (6 µM) heat denaturation at 55 °C for 50 min, determined by light scattering at 360 nm, in the presence of TDC3 (6 µM) having a substitution of a specific amino acid residue: TDC3 (control) (open squares), TDC3^K662L^ (open diamonds), TDC3^L702E^ (closed squares), TDC3^F705E^ (open circles), TDC3^R710W^ (closed triangles), TDC3^F711E^ (closed circles), and TDC3^D714I^ (open triangles). (**B**) Average of the final light-scattering value from three independent experiments. (**C**) ADH heat denaturation rate in the presence of the substitution variants, average of three independent experiments.

## Data Availability

Not applicable.

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
