# Peer review of "The Molecular Chaperone Mechanism of the C-Terminal Domain of Large-Size Subunit Catalases"

_antioxidants, 2023, doi:10.3390/antiox12040839_

Round 1

Reviewer 1 Report

The manuscript authored by Nava-Ramírez and Wilhelm Hansberg analyze the molecular chaperone function of the CT of CAT-3. The work is cleary described, the materials and methods sections are clearly and in detail presented and the results and discussion the same. 

One of the observation is related to section 3.4 where I wpuld like to ask if the protein structeres were from protein data bank or any shor molecular dynamics studies were performed?

The data from the Figures where no error bars were included (Fig 1a, 5c, 6a) are presented as a medium value of 2/3 experiments or are single measuments? 

The conclusion part can be extended because the amount of experimental data is big. 

Author Response

One of the observation is related to section 3.4 where I wpuld like to ask if the protein structeres were from protein data bank or any shor molecular dynamics studies were performed?

We used the InterProSurf server with our PDB file to predict the zones of the TDC3 dimer that could interact with other proteins. A reference is now given, line 276.

The data from the Figures where no error bars were included (Fig 1a, 5c, 6a) are presented as a medium value of 2/3 experiments or are single measuments? 

Traces in Figure 1A, 5C, and 6A are from one experiment. Mean values and standard deviation are reported in 1B, 5D and 6B. We now include in the legend Figure 1A, 5C, and 6A: "representative traces of..." to indicate that it is one experiment: lines 203, 279, and 307.

The conclusion part can be extended because the amount of experimental data is big. 

We now include two more sentences: "Deletion of this loop resulted in major loss of TDC3 chaperone activity", lines 380 and 381, and "Substitution of hydrophobic for charged amino acid residues and vice versa in this loop decreased considerably the unfolding activity."  lines 382-384.

Reviewer 2 Report

In the manuscript “The molecular chaperone mechanism of the C-terminal domain of large-size subunit catalases” the authors analyze the mechanism of the C-terminal domain (CT) of Large-size subunit catalases (LSCs) as an unfolding enzyme. The authors showed that the dimeric form of catalase-3 (CAT-3) CT (TDC3) of Neurospora crassa has higher activity then the monomeric form. Furthermore, the variant of CAT-3 CT lacking the terminal loop of 17 amino acids lost most of its unfolding activity. Substituting charged for hydrophobic residues in this C-terminal loop diminished the molecular chaperone activity, indicating that these amino acids play a relevant role in its unfolding activity.

The results reported by the authors provide relevant functional insights. I would recommend manuscript publication after addressing the following issues.

Major issues:

1.     In Section 2.4 the authors reports that the monomer and dimer formation was verified by SDS-PAGE. Since this analysis is done under protein denaturing condition, the authors should explain how they use this technique to verify protein folding and quaternary structure.

2.     In Section 3.4 the authors report that they mutated some hydrophobic and hydrophilic residues taking care to not disturb the conformation of the domain, but the methods used to verify the retention of the folding in the mutants are not reported/described. This analysis is important to investigate further the protein activity.

Minor issues:

1.     The keywords “hydrophobic amino acid residues; charged amino acid residues” seem too general, please revise.

2.     Section 2.2. Primer sequences should be specified (in the main text or in the Supporting Information)

3.     Section 2.2 and 2.3. A purification procedure is reported at lines 124-128 of Section 2.2 but protein purification is reported in Section 2.3. Please revise either section content or titles.

4.     Section 2.4. Please explain the meaning of “deionized urea”

5.     Section 2.7, line 173. “heat ADH denaturation assay” could be changed in “heat denaturation assay of ADH”

6.     Section 3.1, line 196. “two-fold” should be changed in “two times”.

7.     Figure 5B is too small and labels are difficult to read, please revise it.

8.     Section 3.4, line 289. “residues do participate” should be changed in “residues participate”

9.     Lines 302-308 are part of the caption of Figure 6.

10.  Section 4, lines 344-348. The sentence is too long, please revise.

11.  Section 4, lines 364-365. “have an antioxidant and an unfolding activities” should be changed in “have antioxidant and unfolding activities”.

12.  English should be carefully revised throughout the manuscript.

Author Response

Major issues:

  1. In Section 2.4 the authors reports that the monomer and dimer formation was verified by SDS-PAGE. Since this analysis is done under protein denaturing condition, the authors should explain how they use this technique to verify protein folding and quaternary structure.
  2. In Section 3.4 the authors report that they mutated some hydrophobic and hydrophilic residues taking care to not disturb the conformation of the domain, but the methods used to verify the retention of the folding in the mutants are not reported/described. This analysis is important to investigate further the protein activity.

Both mayor issues posed by the reviewer address the question whether the TDC3, monomer and dimer, and the mutant variants in the loose terminal end have a proper folding. This is a tough question for which there is no clear answer, unless structural studies (NMR or crystallography) are undertaken.

           If the protein is soluble and retains the activity studied, it is usually assumed that it is properly folded. The TDC3, monomer and dimer, and mutant variants, are soluble proteins that retain at least some chaperone activity. The TDC3 confers stability to the catalase domain and has an amino acid composition of very stable proteins (Hansberg et al. Antioxidants, 2022), which would not be expected to unfold. Besides, the loose C-terminal end, in which some amino acid residues were substituted, does not interfere with the structure of the TDC3. Thus, it is a reasonable assumption that the TDC3, monomer and dimer and substitution variants are properly folded.

           Nevertheless, we now provide Circular Dichroism Spectroscopy data of the TDC3, monomer and dimer, and of three mutant variants. Results indicate that TDC3 is structured with similar amounts of alpha helices and beta strands in all proteins tested. Only TDC3D714I showed an increase in the amounts of helices and a decrease in unstructured regions, suggesting that this substitution variant presents a helix instead of the loose C-terminal coil, which could account for its slow rate in its chaperone activity. These experiments are now stated in the text (lines 151-154, 301-303) and presented in supplementary Figure S2.

"taking care to not disturb the conformation of the domain" was substituted for: "To select substitutions that have a less disturbing effect on the conformation of the domain a theoretical analysis was done using the server Duet [43]." lines 294-295.

Minor issues:

  1. The keywords “hydrophobic amino acid residues; charged amino acid residues” seem too general, please revise.

We changed the two keywords for: "chaperone function of hydrophobic and charged amino acid residues" line 26

  1. Section 2.2. Primer sequences should be specified (in the main text or in the Supporting Information)

            Primary sequence is now included in Figure S2. 

  1. Section 2.2 and 2.3. A purification procedure is reported at lines 124-128 of Section 2.2 but protein purification is reported in Section 2.3. Please revise either section content or titles.

Last paragraph was moved to 2.4 and the title of this section was changed to: "TDC3 analysis and TDC3 dimer and monomer formation". Page 4.

  1. Section 2.4. Please explain the meaning of “deionized urea”

Deionized urea is a solution of urea that is passed through a resin to eliminate the degradation products ammonium ion and carbonic acid. However instead of using deionized urea we actually used HPLC grade urea. This is now indicated, line 141.

  1. Section 2.7, line 173. “heat ADH denaturation assay” could be changed in “heat denaturation assay of ADH”

Done, line 175.

  1. Section 3.1, line 196. “two-fold” should be changed in “two times”.

Changed, line 198.

  1. Figure 5B is too small and labels are difficult to read, please revise it.

Figure 5B has been amplified to show more clearly the labeled residues. Page 9.

  1. Section 3.4, line 289. “residues do participate” should be changed in “residues participate”

Done, line 292.

  1. Lines 302-308 are part of the caption of Figure 6.

It is now corrected.

  1. Section 4, lines 344-348. The sentence is too long, please revise.

It was divided in two sentences, lines 351 - 353.

  1. Section 4, lines 364-365. “have an antioxidant and an unfolding activities” should be changed in “have antioxidant and unfolding activities”.

Changed, lines 373 and 374.

  1. English should be carefully revised throughout the manuscript.

            English was revised.

Round 2

Reviewer 2 Report

The authors have made efforts in improving the manuscript. I support manuscript acceptance in the current form.